# DeepIOD: Towards A Context-Aware Indoor–Outdoor Detection Framework Using Smartphone Sensors

**DOI:** 10.3390/s24165125

**Published:** 2024-08-07

**Authors:** Muhammad Bilal Akram Dastagir, Omer Tariq, Dongsoo Han

**Affiliations:** Korea Advanced Institute of Science and Technology—KAIST, Daejeon 34141, Republic of Korea; bilal@kaist.ac.kr (M.B.A.D.); omertariq@kaist.ac.kr (O.T.)

**Keywords:** context awareness, indoor–outdoor detection, location-based services, deep learning, majority voters, smartphone sensors

## Abstract

Accurate indoor–outdoor detection (IOD) is essential for location-based services, context-aware computing, and mobile applications, as it enhances service relevance and precision. However, traditional IOD methods, which rely only on GPS data, often fail in indoor environments due to signal obstructions, while IMU data are unreliable on unseen data in real-time applications due to reduced generalizability. This study addresses this research gap by introducing the DeepIOD framework, which leverages IMU sensor data, GPS, and light information to accurately classify environments as indoor or outdoor. The framework preprocesses input data and employs multiple deep neural network models, combining outputs using an adaptive majority voting mechanism to ensure robust and reliable predictions. Experimental results evaluated on six unseen environments using a smartphone demonstrate that DeepIOD achieves significantly higher accuracy than methods using only IMU sensors. Our DeepIOD system achieves a remarkable accuracy rate of 98–99% with a transition time of less than 10 ms. This research concludes that DeepIOD offers a robust and reliable solution for indoor–outdoor classification with high generalizability, highlighting the importance of integrating diverse data sources to improve location-based services and other applications requiring precise environmental context awareness.

## 1. Introduction

In recent years, indoor–outdoor detection (IOD) has experienced significant growth due to its application in positioning technologies and environmental change detection using multimodal smartphone sensors. IOD plays a crucial role in enhancing the deployment of location-based services in embedded systems characterized by low power consumption and the use of artificial intelligence on the device. These advances offer substantial economic and technical benefits, particularly in the development of integrated indoor–outdoor GPS systems, as illustrated in Figure 1 [1,2,3].

The versatility of IOD extends to numerous applications, including pedestrian location and movement tracking [4,5,6,7,8], activity recognition [9,10], the classification of transportation modes [11,12,13], power management, and medical care [14]. It is integral to the seamless implementation of positioning and navigation systems, effectively bridging indoor and outdoor localization [15,16]. IOD and contextual factors such as time and weather enable personalized services such as adjustable screen brightness and volume according to environmental conditions [17]. IOD models leverage smartphone sensor data to discern a user’s environment, with studies highlighting the behavior of context-aware sensors that exhibit distinct patterns indoors and outdoors. Analyzing these attributes facilitates the prediction of user behavior in various settings. For example, signal strengths typically weaken as users traverse indoor spaces such as doors, stairs, or elevators, where lighting and magnetic fields fluctuate [18,19].

Previously, researchers have pursued multiple methodologies for IOD, including threshold-based approaches based on preset sensor values [20,21]. Zhou et al. (2012) and Ali et al. (2018) discuss threshold-based approaches that rely on preset sensor values [20,21]. Saffar et al. (2019) [22], Zhu, Y. et al. (2019) [17], and Zhu, F. et al. (2023) [23] propose machine learning models that adapt to diverse data through detailed feature extraction. However, these methods often face challenges regarding temporal variations and computational complexity. Thus, deep learning models for temporal elements have been studied, including those by Zhu, Y et al. (2021) [24]. For example, IOD is treated as a time series classification (TSC) problem by Hamideche et al. (2022) [25], while Bakirtzis et al. (2022)  [26] consider it a multivariate TSC utilizing deep learning with self-attention mechanisms and spatial pyramids. Although these approaches achieve high accuracy, they encounter challenges such as a disruption of temporal relationships due to data randomization and the high dimensionality of deep learning TSC data, which require extensive feature engineering and affect training time. Tamborini et al. (2018) [27] and Malik et al. (2023) [28] discuss these challenges.

Despite considerable advances in sensor fusion and machine learning techniques for context detection, several research gaps persist. Traditional IOD methods, highly dependent on GPS data, often fail in indoor environments due to signal obstructions [20], while data from the Inertial Measurement Unit (IMU) are not reliable for unseen data in real-time applications due to reduced generalizability [29]. In addition, there is an inadequate integration of behavioral patterns into context detection models, an underutilization of smartphone sensor data for environmental scene detection, and inefficiencies in classifier combination techniques. Existing models often fail to fully integrate environmental characteristics and carrier behaviors, resulting in imprecise indoor–outdoor context detection [23].

A significant challenge remains in the interpretability of deep learning algorithms used in IOD models. Although these algorithms improve accuracy and efficiency, their complexity hinders the understanding of the rationale behind predictions, complicating the identification and correction of errors. Enhancing the interpretability of these models is essential to improve their transparency, trustworthiness, and practical applicability in real-world IoT environments [26]. Addressing these research gaps will facilitate the development of more effective and energy-efficient IOD models, enhancing navigation capabilities across diverse spatial environments.

This paper proposes the DeepIOD framework, which integrates data from IMU sensors, GPS, and light sensors. The framework preprocesses these data and utilizes multiple deep neural network models and sensor modules to predict whether the environment is indoor or outdoor. The decision is refined using an adaptive majority voter system that considers inputs from all components to ensure accurate classification. This hybrid approach ensures reliable IOD even in challenging scenarios and unseen environments. We evaluated the framework on readily available Android devices by developing the DeepIOD App, which integrates TensorFlow Lite models for the edge implementation of our DNN models. These results demonstrate the superiority of our approach over existing methods, paving the way for more reliable and efficient IOD in smart IoT environments.

The contributions of this work can be summarized as follows:We propose the DeepIOD framework, which accurately classifies environments as indoor or outdoor by dynamically adjusting thresholds based on environmental data, including GPS, light, and other spatiotemporal information. This framework integrates deep neural network (DNN) models utilizing IMU sensor data, with their outputs combined using an adaptive majority voting mechanism, ensuring robust and reliable predictions.We propose a novel voting classifier for indoor–outdoor detection that addresses limitations in traditional methods like Plurality, STV, and Condorcet. This classifier ensures a more robust and transparent decision-making process by integrating pairwise comparisons, overall rankings, and systematic tie-breaking mechanisms, enhancing accuracy and fairness in complex scenarios.We developed the DeepIOD Android application, which operates in real-time, enabling seamless integration with various applications that require timely indoor–outdoor detection capabilities.Extensive experiments conducted on Android devices demonstrate the efficacy of DeepIOD, with accuracy rates ranging from 98% to 99%. These results surpass existing methods based on thresholding, traditional machine learning, and shallow/deep learning techniques.

This paper is organized as follows: Section 2 provides a comprehensive review of related work in indoor–outdoor detection. Section 3 presents the methodology and architecture of DeepIOD in detail. Section 4 discusses the experimental setup and presents the performance evaluation results. Finally, Section 5 concludes the paper with a summary of the contributions and outlines directions for future research.

## 2. Related Works

Several IOD systems have been proposed in recent years, categorized into three main types: (1) algorithms based on multisensor fusion, (2) algorithms based on time-dependent models, and (3) algorithms based on behavioral association.

### 2.1. Algorithms Based on Multisensor Fusion

Single-sensor-based IOD models have inherent limitations in adequately characterizing the environment. To address this, efforts have been made to take advantage of large amounts of feedback from various smartphone sensors. For example, IODetector [20] used lightweight sensors such as a light sensor, magnetometer, and cellular signal to detect indoor–outdoor switching without prior assumptions. SenseIO [21] used measurements from sensor-rich smartphones to infer fine-grained environment types. NeuralIO [30] fusion data from various sensors throughout a neural network were used to determine the indoor–outdoor status. Zhu et al. [24] proposed a multisensor model for indoor–outdoor switching detection using multiple neural network modules. Zeng et al. [31] combined sensors such as the light sensor, magnetic sensor, and GNSS to improve position accuracy in indoor–outdoor scenes. Other models, such as those by Li et al. [1], Radu et al. [32], and Anagnostopoulos et al. [33], also focused on accurate and energy-efficient indoor/outdoor switching detection using different sensor combinations.

### 2.2. Algorithms Based on Time-Dependent Models

Time-dependent IOD models consider the instant characteristics of navigation context features and their connectivity between successive epochs, which is crucial in detecting scene switching. Effective time-dependent machine learning (ML) and deep learning (DL) models have been developed for indoor–outdoor scene detection. For example, Gao et al. [34] used a probabilistic support vector machine (SVM) followed by an HMM (SVM-HMM) to detect scenes with GNSS measurements, improving detection performance through time-domain filtering. Zhu et al. [17] introduced a novel IOD method that filters primary prediction results from different ML classifiers with HMM and integrates them using an ensemble model, achieving high recognition accuracy. The recognition accuracy exceeded 80% for switching delays within 4 s. Xia et al. [35] employed Recurrent Neural Networks (RNN) to address scene switching, achieving a recognition accuracy of 90.94% with a maximum transition delay of 3 s. Zhu et al. [24] introduced a multisensor fusion model that uses DenseNet to extract high-level features and Long Short-Term Memory (LSTM) to capture temporal sequence patterns. Recently, Bakirtzis et al. [26] explored the effectiveness of different deep learning (DL) architectures for indoor–outdoor detection and proposed a model using self-attention mechanisms and incorporating spatial pyramids.

### 2.3. Algorithm Based on Behavior Association

MobiIO [36] introduced a lightweight IOD architecture based on probabilities of human motion activities in indoor and outdoor scenes generated by an SVM. Subsequently, a Hidden Markov Model (HMM) was utilized to estimate the most probable state of the environmental context. Gao et al. [34] successfully implemented behavioral association in indoor–outdoor and foot/vehicle scenes, involving adjustments to SVM-HMM state transition probabilities and associations between environment and behavior labels, termed B-SVM-HMM. This approach not only allows one to revise the parameters of the detection model based on behavioral probabilities, but also facilitates the analysis of differences in environmental characteristics between behavior categories. Recently, Zhu et al. [23] used context connectivity and behavior association to accurately detect environment scenes using a smartphone multisensor fusion. This innovative approach allows for seamless navigation across outdoor, semi-outdoor, and indoor spaces with low energy consumption.

Previous research has explored various indoor/outdoor context detection methods, including sensor-based support vector machine (SVM) and hidden Markov models (HMM). While these approaches have shown promise, they often do not fully exploit the potential of smartphone sensors for context detection. In addition, classifier combination techniques have been used to integrate sensor data, but there is still room for improvement to achieve higher accuracy and energy efficiency. Despite advances in sensor fusion and machine learning techniques for context detection, several research gaps remain unaddressed. The current research landscape reveals several critical gaps in context detection models. Firstly, there is a limited integration of behavior association within these models, which impedes their ability to interpret contextual information accurately. Secondly, the utilization of smartphone sensor data for detecting environmental scenes remains inadequate, further limiting the effectiveness of these models. Additionally, there is a noticeable deficiency in efficient classifier combination techniques, which are essential for maximizing the strengths of individual sensor-based models. Consequently, there is a pressing need to develop a comprehensive model that can accurately detect indoor and outdoor contexts by considering both environmental characteristics and carrier behaviors. Therefore, by bridging these research gaps, a more effective and energy-efficient indoor/outdoor context detection model can be developed, offering enhanced navigation capabilities in diverse spatial environments.

## 3. Problem Formulation and Objectives

This study aims to develop a Deep Learning (DL)-based adaptive context-aware indoor–outdoor detection framework for mobile edge computing (MEC). The framework aims to accurately classify the indoor–outdoor contexts of users based on sensory data *D* and location information *L*. The classification error function ε(s,y,X) is defined in Equation (Equation 1) as
(1)$$\varepsilon \left(s,y,X\right)=\frac{1}{N}\sum_{t=1}^{N}\left[y_{t}\neq f\left(X,\theta \right)\right],$$
where $\left[y_{t}\neq f\left(X,\theta \right)\right]$ is the indicator function, defined in Equation (Equation 2) as
(2)$$\left[y_{t}\neq f\left(X,\theta \right)\right]=\left\{\begin{array}{cc} 1, & \mathrm{if}\;y_{t}\neq f\left(X,\theta \right)\\ 0, & \mathrm{if}\;y_{t}=f\left(X,\theta \right) \end{array}\right..$$

In the above equations, *X* represents the input features, including sensory data *D* and location information *L*. Specifically, *D* encompasses data collected from various sensors, such as the IMU sensors (Magnetometer, Accelerometer, Gyroscope), light sensors, and GPS. The location information *L* refers to the GPS coordinates and any additional contextual location data that might influence the indoor–outdoor classification. The function f(X,θ) denotes the deep learning model that takes the combined input features X={D,L} and predicts the context $y_{t}$ (either indoor or outdoor). The parameter θ represents the model parameters learned during the training phase. The indicator function $\left[y_{t}\neq f\left(X,\theta \right)\right]$ checks whether the predicted context matches the actual context, contributing to the overall classification error ε(s,y,X).

The objective of the DeepIOD system is to find the optimal parameter set θ that minimizes the classification error in Equation (Equation 3):(3)$$\theta^{*}=\arg\;\min_{\theta \in \Theta }\varepsilon \left(s,y,X\right),$$
where Θ represents the parameter space for the model.

The probability of correctly detecting the class $c^{*}$ (indoor or outdoor) among *m* classes (in this case, indoor or outdoor), with a profile of *n* classifiers, each one with accuracy p∈[0,1], using the proposed voting classifier, is given in Equation (Equation 4) by
(4)$$T\left(p\right)=\frac{1}{K}\sum_{i=1}^{\left\lceil \frac{m}{n}\right\rceil }\phi_{i}\cdot p^{i}\cdot {\left(1-p\right)}^{n-i},$$
where $\phi_{i}$ is defined as the coefficient of the monomial $x^{n-i}$ in the expansion of the generating function in Equation (Equation 5):(5)$$G_{m}^{i}\left(x\right)={\left(\sum_{j=0}^{i}\frac{x^{j}}{j!}\right)}^{m-1}.$$

*K* is a normalization constant, defined in Equation (Equation 6) as
(6)K=∑j=0nnjpj(m−−1)n−j(1−−p)n−j.

The framework incorporates an adaptive threshold mechanism for light intensity (T(D)) for varying environmental conditions. This threshold depends on space and time continuity and is crucial for accurate indoor–outdoor detection. Let I(t,x) represent the intensity of light at time *t* and location x. The adaptive threshold function can be formulated in Equation (Equation 7) as
(7)T(D)=adaptiveThreshold(I(t,x)).

Additionally, a fixed threshold ($T_{GPS}$) is utilized for GPS data to determine the indoor–outdoor context. Let GPS(t) denote the GPS coordinates at time *t*. The fixed threshold can be formulated in Equation (Equation 8) as
(8)$$T_{GPS}=\mathrm{fixedThreshold}\left(\mathrm{GPS}\left(t\right)\right).$$

An adaptive majority voter is employed for the final classification decision, which combines the outputs of different models based on weights. The objective of the adaptive majority voter is to determine the predicted context $\hat{C}$ based on the weighted sum of probabilities assigned by each DL model. This can be formulated in Equation (Equation 9) as
(9)$$\hat{C}=\underset{j}{\mathrm{argmax}}\sum_{i=1}^{n}w_{i}p_{i,j},$$
where $\hat{C}$ is the predicted context, *j* indexes the possible contexts, $w_{i}$ represents the weight assigned to module $M_{i}$, and $p_{i,j}$ is the probability assigned to context *j* by module $M_{i}$.

The overall objective of the framework is to accurately classify users’ indoor–outdoor contexts by integrating sensory data, location information, adaptive threshold mechanisms for light intensity and GPS, and social choice mechanisms for combining decisions from DL models. This facilitates robust and context-aware indoor–outdoor detection in mobile environments, which is essential for providing personalized and location-based services.

## 4. Methodology

In this section, we present a DeepIOD framework, an integrated system designed to determine indoor or outdoor environments based on a combination of threshold and deep neural network (DNN) models using smartphone sensors, as shown in Figure 2. The DeepIOD framework addresses the intervention of multiple signals by integrating data from various smartphone sensors and processing them through a combination of deep neural network models and an adaptive majority voting mechanism. This multi-signal approach ensures accurate indoor and outdoor environmental classification by leveraging each sensor type’s strengths. Here is how the framework effectively handles multiple signals. Firstly, the framework collects data from multiple sensors, including IMU (Inertial Measurement Unit), light sensors, and GPS. The collected data undergo preprocessing to extract relevant features and normalize the values, making them suitable for deep neural network (DNN) models. These models (Model A, B, and C) independently analyze the preprocessed IMU data and each produces its prediction regarding the environment. The predictions from these models are then aggregated using a weighted majority rule. The voting system assigns weights to the predictions based on each model’s performance, ensuring that more reliable models have a greater influence on the final decision. In addition to the DNN models, light sensor data are processed using an adaptive threshold approach to evaluate ambient light levels, aiding in the differentiation between indoor and outdoor environments. GPS data are also processed to provide additional location context. Specifically, the horizontal accuracy of the GPS signal is used to distinguish between indoor and outdoor environments, with a threshold of 5 m typically used as a criterion [37].

The Adaptive Majority Voter (AMV) mechanism consolidates the output of the DNN models, the light module, and the GPS module. It uses a weighted sum approach to combine these inputs, dynamically adjusting the weights based on the reliability and performance of each model and sensor data. This approach ensures that the most reliable predictions have a greater influence on the final decision, resulting in a robust and adaptive classification system. The adaptive weighting scheme allows the system to better handle variations in model performance due to changes in environmental conditions, thereby enhancing the robustness and accuracy of the classification process. By integrating and processing multiple signals, the DeepIOD framework significantly improves its ability to accurately classify indoor and outdoor environments. The combination of diverse sensory data mitigates the limitations of relying on a single type of data, enhancing overall accuracy. The adaptive majority voting mechanism further improves robustness by dynamically adjusting to prioritize the most reliable sources of predictions, ensuring high performance even under varying environmental conditions. In the following section, we will explain the components in more detail.

### 4.1. Proposed DNN Datasream

In the context of indoor–outdoor detection (IOD) using smartphone sensors, we define the data stream and procedures involved in the proposed DeepIOD system shown in Figure 3 as follows.

Let *s* represent the raw signals obtained from a smartphone’s embedded inertial measurement unit (IMU) sensors, comprising nine dimensions, including acceleration, magnetism, and angular velocity. These signals *s* are segmented into *M* segments, denoted as $s_{p}$, where p=1,2,…,M, using the sliding window method. Each segment $s_{p}$ is labeled in a supervised manner as $\left\{s_{p},y_{p}\right\}$, where $y_{p}$ represents the corresponding indoor or outdoor environment label. A data compression model is applied to remove similar segments with the same labels to reduce redundancy and improve efficiency. The resulting compressed dataset $\left\{s_{i},y_{i}\right\}$, where i=1,2,…,N (typically with N≤M), undergoes signal preprocessing for information enhancement, resulting in processed signals $s_{i}^{*}$. The proposed DeepIOD system constructs a classifier f(X,θ) based on a deep learning (DL) algorithm. The input *X* to the classifier can be the extracted features $F_{hs^{*}}$ or the processed signals $s^{*}$. Furthermore, the compressed dataset $\left\{s_{i}^{*},y_{i}\right\}$ is reconstructed as a basic dataset $s_{qk}^{*}$, where k=1,2,…,nc (with nc being the number of classes) and $q\in R^{+}$, for comparison with new activities. During the testing procedure, the classifier predicts the current environment ${\hat{y}}_{t}$ based on the extracted feature $F_{hs_{t}^{*}}$ (based on ML) or the processed signal $s_{t}^{*}$ (based on DL). The classification error ε is calculated using a supervised learning strategy by comparing ${\hat{y}}_{t}$ with the real label $y_{t}$, as shown in Equation (Equation 10).
(10)$$\varepsilon \left(s,y,X\right)=\frac{1}{N}\sum_{t=1}^{N}\left[y_{t}\neq f\left(X,\theta \right)\right],$$
where $\left[y_{t}\neq f\left(X,\theta \right)\right]=\left\{\begin{array}{cc} 1, & \mathrm{if}\;y_{t}\neq f\left(X,\theta \right)\\ 0, & \mathrm{if}\;y_{t}=f\left(X,\theta \right) \end{array}\right.$. The DeepIOD system aims to find the optimal parameter set θ that minimizes the classification error, as represented by Equation (Equation 11):(11)$$\arg\;\min_{\theta \in \Theta }\varepsilon \left(s,y,X\right).$$

During the update procedure, the newly processed input $s_{t}^{*}$ is compared with the training dataset $s_{q}^{*}{\hat{y}}_{t}$ by calculating the correlation coefficient $\rho =\mathrm{corr}(s_{t}^{*},s_{q}^{*}{\hat{y}}_{t})$. If ρ<η (where η=0.8), indicating a significant deviation from the existing training data, the DeepIOD system retrains the classifier on the updated training dataset $\left\{\left[s_{i}^{*};s_{t}^{*}\right],\left[y_{i};nc+1\right]\right\}$, as shown in the Algorithm 1. Through these procedures, the DeepIOD system aims to achieve robust and accurate indoor–outdoor detection using IMU data, adapting to environmental changes and continuously improving its performance.
**Algorithm 1:** Downstream algorithm.
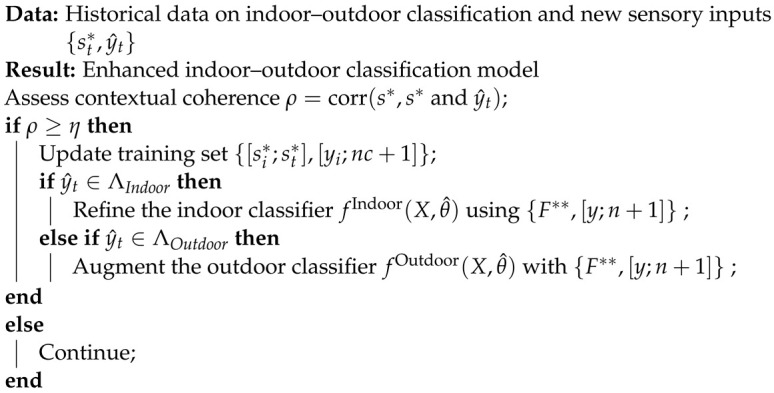


### 4.2. Proposed Models for DNN Downstream

In this section, three innovative models are presented to enhance sequential data analysis in sequential datasets, and the training algorithm is shown in Algorithm 2.
**Algorithm 2:** Training algorithm with cross-entropy loss.
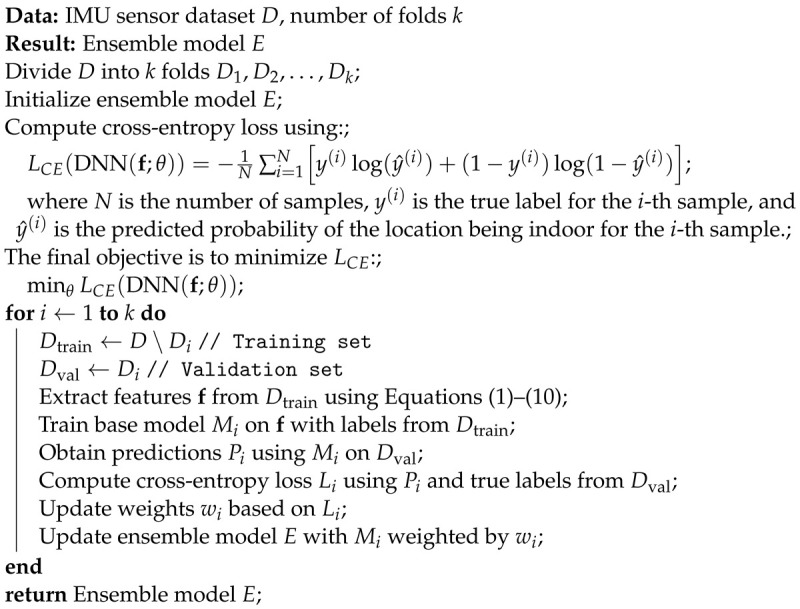


#### 4.2.1. Model A: CNN-LSTM-Based

Model A represents a sophisticated blend of Convolutional Neural Networks (CNNs) and Long Short-Term Memory (LSTM) networks designed to extract rich features from sequential data. This architecture encompasses distinct phases: an encoder phase responsible for processing input sequences and a decoder phase for reconstructing the output representation.

The encoder phase begins with a sequence of convolutional operations characterized by utilizing one-dimensional convolutional layers, followed by rectified linear unit (ReLU) activation functions. These operations can be mathematically represented as
(12)$$f^{\left(i\right)}=\mathrm{ReLU}\left(W^{\left(i\right)}*x+b^{\left(i\right)}\right)$$
where $f^{\left(i\right)}$ is the output feature map of the *i*-th convolutional layer, $W^{\left(i\right)}$ is the filter, x is the input, and $b^{\left(i\right)}$ is the bias term. Subsequently, the output is downsampled through max-pooling operations to reduce spatial dimensions while retaining essential features:(13)$$p^{\left(i\right)}=\mathrm{MaxPool}\left(f^{\left(i\right)}\right)$$

The resultant feature maps are then fed into LSTM units to capture temporal dependencies within the data:(14)$$h_{t}=\mathrm{LSTM}\left(p^{\left(i\right)},h_{t-1}\right)$$
where $h_{t}$ represents the hidden state at time step *t*. In the decoder phase, the encoded representations are further processed through convolutional layers and max-pooling operations to extract higher-level features. The decoder employs LSTM units similarly to refine the temporal information encoded in the feature maps:(15)$$h_{t}^{\prime }=\mathrm{LSTM}\left(p^{\left(i\right)},h_{t-1}^{\prime }\right)$$

Finally, the outputs from the encoder and decoder branches are concatenated to form the final representation:(16)$$y=\mathrm{Concat}(h_{t},h_{t}^{\prime })$$

This concatenated output is then processed through dropout and softmax layers for the final classification:(17)$$y_{\mathrm{final}}=\mathrm{Softmax}\left(\mathrm{Dropout}\left(y\right)\right)$$

#### 4.2.2. Model B: CNN-MHA-Based

Model B introduces a novel architecture designed to analyze sequential data using Convolutional Neural Networks (CNNs) and Multi-Head Attention (MHA) mechanisms. This model comprises encoder and decoder operations tailored to process input sequences and reconstruct the output representation efficiently. In the encoder phase, the model begins with a series of convolutional layers:(18)$$f^{\left(i\right)}=\mathrm{ReLU}\left(W^{\left(i\right)}*x+b^{\left(i\right)}\right)$$

Layer normalization is then applied to stabilize the training:(19)$$f_{\mathrm{norm}}^{\left(i\right)}=\mathrm{LayerNorm}\left(f^{\left(i\right)}\right)$$

Multi-head self-attention mechanisms are subsequently employed to extract hierarchical features from the input sequence:(20)$$z^{\left(i\right)}=\mathrm{MHA}\left(f_{\mathrm{norm}}^{\left(i\right)}\right)$$

The resulting feature maps are reshaped and aggregated to form a compact representation:(21)$$z_{\mathrm{agg}}=\mathrm{Aggregate}\left(z^{\left(i\right)}\right)$$

In the decoder phase, these encoded representations are further refined through additional convolutional layers and self-attention mechanisms:(22)$$h_{t}=\mathrm{Conv}\left(z_{\mathrm{agg}}\right)+\mathrm{MHA}\left(z_{\mathrm{agg}}\right)$$

This iterative process enables the model to reconstruct the output sequence while preserving essential spatial and temporal information. Finally, the outputs from the encoder and decoder branches are concatenated:(23)$$y=\mathrm{Concat}(h_{t},z_{\mathrm{agg}})$$

#### 4.2.3. Model C: Depth-Wise Separable Convolutional Neural Network

Model C capitalizes on the efficiency of depth-wise separable convolutions for learning spatial and channel-wise features within sequential data. The network employs depth-wise convolutions followed by point-wise convolutions to efficiently learn spatial and channel-wise features. Depth-wise convolutions are performed as follows:(24)$$f_{\mathrm{depth}}^{\left(i\right)}=\mathrm{ReLU}\left(W_{\mathrm{depth}}^{\left(i\right)}*x+b_{\mathrm{depth}}^{\left(i\right)}\right)$$

Point-wise convolutions are then applied to combine these features:(25)$$f_{\mathrm{point}}^{\left(i\right)}=\mathrm{ReLU}\left(W_{\mathrm{point}}^{\left(i\right)}\cdot f_{\mathrm{depth}}^{\left(i\right)}+b_{\mathrm{point}}^{\left(i\right)}\right)$$

Batch normalization is applied after each point-wise convolution to stabilize learning:(26)$$f_{\mathrm{bn}}^{\left(i\right)}=\mathrm{BatchNorm}\left(f_{\mathrm{point}}^{\left(i\right)}\right)$$

This sequence of depth-wise convolution, point-wise convolution, and batch normalization is repeated, followed by max pooling and dropout to reduce overfitting and dimensionality:(27)$$p^{\left(i\right)}=\mathrm{MaxPool}\left(\mathrm{Dropout}\left(f_{\mathrm{bn}}^{\left(i\right)}\right)\right)$$

As the network deepens, the number of channels is increased to capture more complex features. The final feature maps are flattened and passed through dense layers for classification:(28)$$y=\mathrm{Softmax}(\mathrm{Dense}\left(\mathrm{Flatten}\left(p^{\left(i\right)}\right)\right))$$

Each proposed model offers a unique approach to different architectures, presenting promising avenues for enhanced sequential data analysis. By leveraging the complementary strengths of Convolutional Neural Networks, Depth-Wise Separable Convolutional Networks, LSTMs, and Multi-Head Attention, these models exhibit the potential to extract informative features from diverse sequential datasets. The structural details of each model used in the proposed framework are shown in Table 1.

### 4.3. Propose Voting Classifier

The proposed voting classifier for indoor–outdoor detection offers a novel approach to decision-making, addressing the limitations inherent in traditional voting methods such as Plurality, STV, Condorcet, Borda, Copeland, and Dictatorship as shown in Table 2. This classifier aims to provide a more robust and transparent means of selecting the most preferred option by integrating pairwise comparisons, overall rankings, and systematic tie-breaking mechanisms. Unlike traditional methods, vulnerable to strategic voting or manipulation, the proposed classifier offers a systematic approach that prioritizes the collective preferences of multiple classifiers, ensuring fair and accurate decision-making in complex scenarios such as indoor–outdoor detection. The plurality method is vulnerable to vote splitting and does not ensure the selection of the most preferred option. To mitigate these issues, the proposed improvement suggests considering pairwise comparisons and overall rankings. While allowing for more representative outcomes, the Single Transferable Vote (STV) method suffers from a complex ballot-counting process and is susceptible to strategic voting. The improvement aims to simplify the decision-making process and account for preferences across multiple options. Although robust in certain scenarios, the Condorcet method may not always produce a winner and is vulnerable to cycling. To address these limitations, a systematic approach to tie-breaking is proposed. The Borda count method, known for its susceptibility to strategic manipulation and potential failure to select the most preferred option, can be enhanced by scoring based on pairwise comparisons. The Copeland method faces complexities in determining the winner and is vulnerable to strategic voting. Simplifying the selection process and considering voter preferences are suggested improvements. Lastly, the Dictatorship method, which concentrates power and lacks representation of minority views, can be improved by basing decisions on the collective preferences of multiple classifiers. These proposed improvements aim to enhance the fairness and efficiency of voting systems.
**Theorem** **1.***The probability of correctly detecting the class $c^{*}$ (indoor or outdoor) among m classes (in this case, indoor or outdoor) with a profile of n classifiers, each one with accuracy p∈[0,1], using the proposed voting classifier is given by:*(29)$$T\left(p\right)=\frac{1}{K}\sum_{i=1}^{\left\lceil \frac{m}{n}\right\rceil }\phi_{i}\cdot p^{i}\cdot {\left(1-p\right)}^{n-i},$$*where $\phi_{i}$ is defined as the coefficient of the monomial $x^{n-i}$ in the expansion of the generating function:*(30)$$G_{m}^{i}\left(x\right)={\left(\sum_{j=0}^{i}\frac{x^{j}}{j!}\right)}^{m-1},$$*and K is a normalization constant, defined as*(31)K=∑j=0nnjpj(m−−1)n−j(1−−p)n−j.
**Proof** **of** **Theorem** **1.** Let us break down the derivation of T(p) step by step:
Pairwise Comparison: For each pair of alternatives (indoor or outdoor), the voting classifier assigns scores based on the accuracy of each classifier. After pairwise comparison with another class, the score assigned to a class is computed using the accuracy of the classifiers that favor each class. This process is repeated for all pairwise comparisons in the profile.Calculation of Cumulative Score: After conducting pairwise comparisons, the voting classifier calculates the cumulative score $C_{i}$ for each class *i* by summing up the scores obtained from all pairwise comparisons involving that class. The cumulative score represents the profile’s overall preference or support for each class.Selection of Winners: The class(es) with the highest cumulative score(s) are identified as potential winners. If there is a tie, i.e., if multiple classes share the highest cumulative score, the method proceeds to the next step to break the tie.Overall Ranking: In case of a tie, the voting classifier examines the overall ranking of tied classes within the preference profile. It selects the class that appears most frequently or at the highest positions in the preference profile. The overall ranking $R_{i}$ of each class *i* is determined based on its frequency or position in the preference profile.Resolution of a Tie: If a tie persists even after considering the overall ranking, the voting classifier may break the tie arbitrarily or based on additional criteria specified by the voting context.Computation of T(p):T(p) represents the probability of correctly detecting the class $c^{*}$ (indoor or outdoor) given the accuracy *p* of each classifier in the profile. The formula for T(p) sums up the probabilities of all possible profiles’ votes that correctly detect $c^{*}$, considering varying numbers of classifiers voting for $c^{*}$ and the remaining classifiers voting for other classes. The binomial factor accounts for the number of possible positions of classifiers voting for $c^{*}$, while the probabilities $p^{i}$ and ${\left(1-p\right)}^{n-i}$ represent the likelihood of classifiers voting correctly or incorrectly, respectively. The normalization constant *K* ensures that the probabilities sum up to 1 across all possible profiles.The modified equation
(32)$$x^{j}{\left(\sum_{j=0}^{B}\frac{x^{j}}{j!}\right)}^{A}\overset{A=m-1}{\underset{B=i-1}{\to }}{\left(\sum_{j=0}^{i-1}\frac{x^{j}}{j!}\right)}^{m-1}=G_{i}^{m}\left(x\right)$$
denotes the generating function $G_{i}^{m}\left(x\right)$ used in the proposed voting classifier for indoor-outdoor detection. It encapsulates the probabilities associated with different combinations of classifiers voting for the correct class $c^{*}$ and the remaining classifiers voting for other classes. By applying this equation, we can effectively compute T(p) and determine the probability of accurately detecting the class $c^{*}$ given the accuracy of each classifier in the profile.In the above Equation (Equation 32), the generating function $G_{i}^{m}\left(x\right)$ used in the proposed voting classifier for indoor-outdoor detection encapsulates the probabilities associated with different combinations of classifiers voting for the correct class $c^{*}$ and the remaining classifiers voting for other classes. By applying this equation, we can effectively compute T(p) and determine the probability of accurately detecting the class $c^{*}$ given the accuracy of each classifier in the profile. The theoretical analysis of Theorem 1 is in Appendix A.    □

In Algorithm 3, the term ‘profile’ refers to the dataset comprising pairwise comparisons between alternatives. Initially, the algorithm initializes a dictionary, δ[α], to maintain the scores of each alternative, where α denotes individual alternatives. It then iterates through each preference in the profile, evaluating each pair of alternatives, denoted as ζ and ξ. For each distinct pair, the algorithm updates the scores accordingly, incrementing the score of ζ and decrementing the score of ξ. After processing all preferences, it determines the maximum score, denoted as ϵ, among all alternatives. Subsequently, it identifies a set, ι, containing alternatives with the maximum score. If multiple alternatives exist in ι, Algorithm 4 calculates their overall rankings using a designated function, “calcOverallRank”, and selects the alternative with the highest overall ranking. However, if there is only one alternative in ι, it directly designates it as the winning alternative and concludes the algorithm by returning the selected alternative.

In the “Function to Calculate Overall Ranking”, the algorithm aims to determine the overall ranking of a specific alternative, denoted as α, within the given profile dataset. It initializes a variable, ρ, to accumulate the overall ranking of α. The algorithm then iterates through each preference in the profile, denoted as ϕ, assessing whether the alternative α appears in each preference. If α is present in a preference, the algorithm adds its position or index to ρ. Once it has traversed all preferences, the algorithm returns the accumulated value of ρ as the overall ranking of the alternative α.

**Algorithm 3:** Propose voting classifier algorithm.

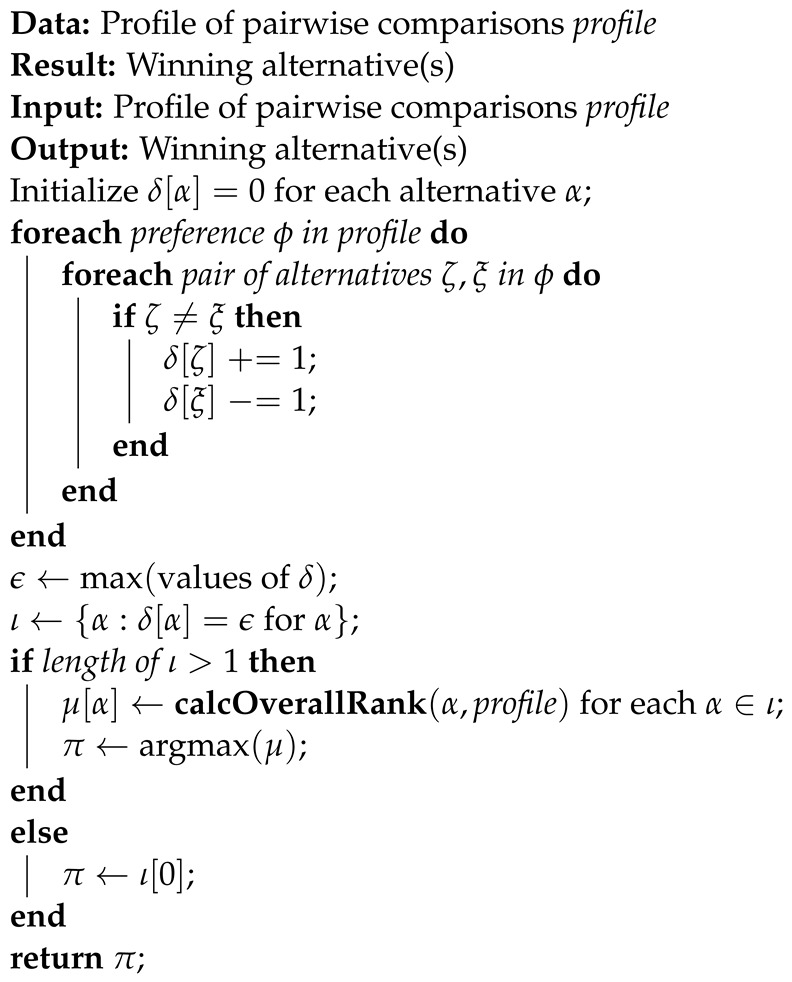



**Algorithm 4:** Function to calculate overall ranking.

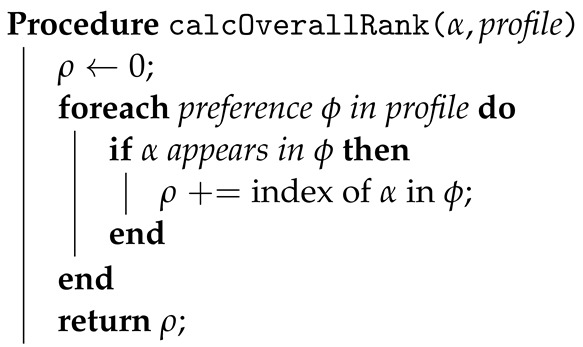



### 4.4. Adaptive Threshold Mechanism for Light Intensity

To accurately detect indoor and outdoor contexts under varying environmental conditions, we have developed an adaptive threshold mechanism for light intensity, denoted as T(D). This mechanism dynamically adjusts the threshold based on space and time continuity, thereby improving the robustness of the classification system. The intensity of light at a given time *t* and location x, represented as I(t,x), forms the basis of this adaptive threshold. The function for the adaptive threshold, T(D), is defined and shown in Equation (Equation 33):(33)T(D)=adaptiveThreshold(I(t,x)).

This formulation ensures that the light intensity threshold adapts to real-time changes in the environment, mitigating the effects of transient lighting variations and enhancing the reliability of context detection. The threshold for light intensity in the framework is adaptively adjusted by continuously monitoring the light intensity I(t,x) at different times and locations. The adaptive threshold function adaptiveThreshold(I(t,x)) takes into account the dynamic environmental conditions, such as changes in natural lighting due to weather, time of day, and the specific indoor or outdoor location. The key steps for adjusting the threshold are as follows:Continuous Monitoring: The system continuously measures the light intensity I(t,x) at specific time intervals and locations. This provides real-time data reflecting current lighting conditions.Dynamic Adjustment: Using the real-time light intensity data, the adaptive threshold function recalculates T(D). This function is designed to respond to significant changes in light intensity, ensuring that the threshold reflects the current environment accurately.Space and Time Continuity: The adaptive threshold function incorporates principles of space and time continuity. This means it considers the recent history of light intensity measurements and the spatial context (e.g., whether the user is near a window or in a corridor) to make more informed adjustments. This reduces the likelihood of erroneous context detection due to sudden, brief changes in lighting, such as someone turning on a light.Contextual Sensitivity: By being sensitive to the context in which measurements are taken, the adaptive threshold can differentiate between typical indoor lighting patterns and outdoor lighting conditions. For example, even if indoor lighting is bright, the pattern of intensity changes over time can help distinguish it from outdoor light, which typically varies more gradually.

This adaptive approach allows the threshold T(D) to be flexible and responsive, enhancing the system’s ability to accurately classify whether a user is indoors or outdoors under varying lighting conditions. Table 3, below, summarizes the light intensity thresholds for three locations: the Republic of Korea, China, and Pakistan. These thresholds illustrate how the framework adjusts the light intensity threshold adaptively based on the environmental conditions specific to each location.

### 4.5. Fixed Threshold for GPS Data

In addition to the adaptive light intensity threshold, our framework employs a fixed threshold for GPS data to assist in determining indoor and outdoor contexts. Let GPS(t) denote the GPS coordinates at time *t*. The fixed threshold, $T_{GPS}$, is utilized to provide a consistent criterion for classification based on location data. This threshold is formulated as in Equation (Equation 34):(34)$$T_{GPS}=\mathrm{fixedThreshold}\left(\mathrm{GPS}\left(t\right)\right).$$

To classify the GPS signal as indoor or outdoor, horizontal accuracy is used. For Android phones, a horizontal accuracy of greater than 5 m typically indicates an indoor location, while a horizontal accuracy of less than 5 m indicates an outdoor location [37]. Table 4 below summarizes these thresholds.

By incorporating a fixed threshold for GPS data, the system can leverage spatial information to complement light-intensity data, thus providing a more comprehensive basis for context detection.

### 4.6. Adaptive Majority Voter for Classification

We employed an adaptive majority voter mechanism to integrate the outputs from various models and achieve a robust final classification. This mechanism combines the outputs of different models by assigning weights to each model’s prediction. The goal is to determine the predicted context $\hat{C}$ based on the weighted sum of probabilities assigned by each deep learning (DL) model. The objective function for the adaptive majority voter is given by Equation (Equation 35):(35)$$\hat{C}=\underset{j}{\mathrm{argmax}}\sum_{i=1}^{n}w_{i}p_{i,j},$$
where $\hat{C}$ is the predicted context, *j* indexes the possible contexts, $w_{i}$ represents the weight assigned to module $M_{i}$, and $p_{i,j}$ is the probability assigned to context *j* by module $M_{i}$. The adaptive majority voter dynamically adjusts the weights based on each model’s performance, ensuring that the most reliable models have a greater influence on the final decision. This approach leverages the strengths of multiple models and mitigates the impact of any single model’s inaccuracies. Traditional voting methods, such as simple majority voting, treat each model’s output equally, regardless of the model’s performance. This can lead to suboptimal decisions, especially if some models consistently perform better. In contrast, the adaptive majority voting mechanism improves the accuracy of IOD classification by dynamically assigning higher weights to models with better performance. This is achieved through the continuous evaluation of each model’s accuracy over time, allowing the system to adapt and prioritize the most reliable sources of predictions. By incorporating an adaptive weighting scheme, the system can better handle variations in model performance that might occur due to changes in environmental conditions or the inherent complexity of different contexts. As a result, the adaptive majority voter not only enhances the robustness of the classification process but also significantly improves overall accuracy compared to traditional methods. The adaptive majority voting mechanism enhances the accuracy of IOD classification by:Assigning higher weights to more reliable models based on their performance.Dynamically adjusting to changes in model performance over time.Leveraging the strengths of multiple models to mitigate the impact of any single model’s inaccuracies.

This results in a more robust and accurate classification system, capable of adapting to various environmental conditions and maintaining high performance.

### 4.7. Framework Overview

The primary objective of our framework is to accurately classify users’ indoor–outdoor contexts by integrating sensory data, location information, adaptive threshold mechanisms for light intensity and GPS, and social choice mechanisms for combining decisions from DL models. This multi-faceted approach facilitates robust and context-aware indoor–outdoor detection in mobile environments, essential for providing personalized and location-based services. By combining adaptive and fixed thresholds with an adaptive majority voter, our framework effectively addresses the challenges of varying environmental conditions and model uncertainties. This holistic approach ensures high context detection accuracy, enhancing the user experience in mobile applications.

### 4.8. Computational Complexity and Real-Time Performance

Several factors, including the frequency of data sampling, the complexity of the adaptive threshold function, and the overhead of the adaptive majority voter mechanism, influence the computational complexity of the proposed framework. The continuous monitoring of light intensity and GPS data requires frequent sampling, leading to a complexity of O(n), where *n* is the number of samples. Efficient data handling and real-time processing techniques are employed to ensure minimal latency. Data acquisition involves collecting test data from the IMU sensors (Magnetometer, Accelerometer, Gyroscope), light sensor, and GPS, with a complexity of O(n·s), where *s* is the number of sensor types. Preprocessing steps include normalizing IMU data, extracting relevant features, and transforming features into a structured IMU dataset, each with a complexity of O(n·f), where *f* is the number of features per sample. Model evaluation consists of feeding preprocessed IMU data into Model A, Model B, and Model C, and obtaining predictions from each model, resulting in a complexity of O(n·m), where *m* is the number of models. The voting system combines predictions using a weighted majority rule and calculates the final prediction, which involves O(m·c) operations, where *c* is the number of context classes. The light and GPS module processes light data using an adaptive threshold approach with a complexity of O(k), where *k* is the window size for recent measurements, and processes GPS data to determine horizontal accuracy with a complexity of O(n). The adaptive majority voter (AMV) combines the DNN voting system output, light module, and GPS module, resulting in O(m·c) operations. Displaying the final indoor or outdoor classification is an O(1) operation. Overall, the framework’s complexity is O(n·s+n·f+n·m+m·c+k). The system is optimized through efficient data structures and parallel processing to maintain real-time performance. Adaptive mechanisms ensure that the computational load is manageable, allowing timely and accurate context detection without significant delays.

### 4.9. Cost and Memory Complexity

The cost and memory complexity of the framework are critical for ensuring efficient performance on resource-constrained devices such as smartphones. The storage of sensor and intermediate processed data requires memory proportional to the number of samples and features, leading to a memory requirement of O(n·f). Data acquisition also demands memory for collecting raw data from multiple sensor types, resulting in O(n·s) memory complexity. Preprocessing steps require memory for normalized and structured data, with a complexity of O(n·f). Model storage and execution involve a memory complexity of O(m·s), where *s* is the size of each model. Additional memory is needed for model outputs and intermediate computations, adding O(m·c) memory requirement. The adaptive threshold and majority voter mechanisms require memory to store the sliding window of recent data (O(k)) and the weights used for combining model outputs (O(m·c)). Overall, the memory complexity of the framework is O(n·f+n·s+m·s+m·c+k). Optimizations such as efficient data structures, memory reuse, and compression techniques are implemented to minimize the memory footprint and ensure smooth smartphone operation.

### 4.10. DNN Model Conversion to Tensorflow Lite

This section details the lite conversion and integration of deep neural network (DNN) models (A, B, and C) within the DeepIOD framework for the classification of mobile environments, as shown in Figure 4. TensorFlow Lite, a mobile-optimized machine learning framework, facilitates this process. The three DNN models (A, B, and C) were first assessed for compatibility with TensorFlow Lite’s conversion tools. The TensorFlow SavedModel format was ensured for each model to streamline the conversion process. Using the TensorFlow Lite framework tool, each model underwent individual conversion. This conversion converted the models into a mobile-friendly format (.tflite) while maintaining functionality. In addition, the optimization techniques offered by TensorFlow Lite were explored to potentially reduce the model’s size and improve the inference speed on mobile devices. This optimization step aimed to balance model accuracy and efficient resource utilization on the target mobile platform. The converted TensorFlow Lite models (A.tflite, B.tflite, C.tflite) were then incorporated into the DeepIOD framework on the mobile device. The framework’s code was adapted to seamlessly load each converted model using the TensorFlow Lite interpreter API. The DeepIOD framework’s data preprocessing stage was optimized for mobile implementation to prepare sensor data for inference on the mobile device. This involved replicating the mobile device’s core data normalization and feature extraction functionalities. Once the pre-processed data are available, the TensorFlow Lite interpreter executes the inference tasks on each model (A, B, and C) independently. The individual model predictions are then obtained for further processing. The framework implements the weighted majority voting logic on the mobile device. This logic combines the predictions of models A, B, and C to determine the most probable environment classification (indoor or outdoor). Using TensorFlow Lite, the DeepIOD framework achieves efficient machine learning for mobile environment classification on the device. TensorFlow Lite’s optimization techniques contribute to a reduction in model size and an improvement in inference speed. This combination empowers the DeepIOD framework for real-time operation on mobile devices, making it suitable for various applications requiring environmental awareness.

### 4.11. Real-Time DeepIOD App

The DeepIOD app is designed for indoor–outdoor real-time detection using a combination of GPS, Light, Barometer, and IMU sensors and a deep neural network (DNN) model. Algorithm 5 outlines the steps for deploying the framework on an Android smartphone, encompassing data acquisition, preprocessing, model evaluation, and final decision-making through an adaptive majority voter (AMV) mechanism. Figure 5 shows the app in action, correctly identifying whether the user is indoors or outdoors. The app processes sensor data in real-time, leveraging the trained deep learning model to determine the user’s environment, which is then displayed on the screen with a corresponding label (“Outdoor” or “Indoor”).
**Algorithm 5:** Deployment algorithm for indoor–outdoor detection on mobile.
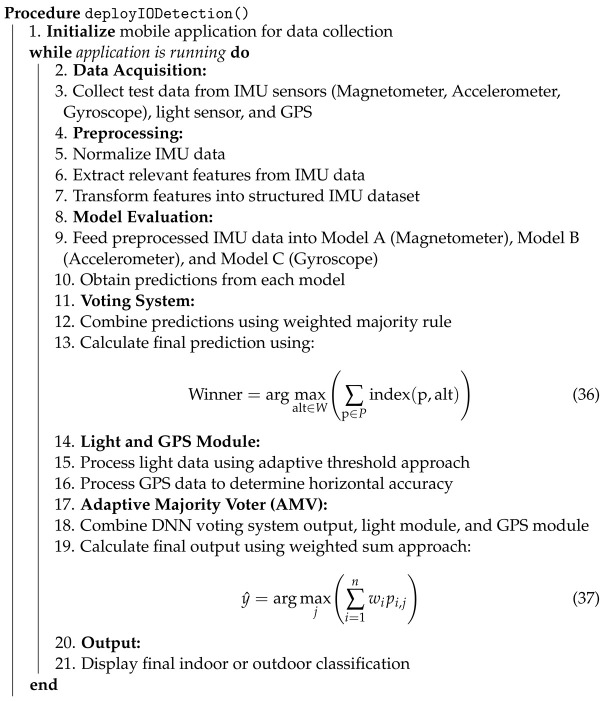


## 5. Framework Implementation and Discussion

In this section, we start by describing IOD datasets, followed by the model’s offline learning, and finally, the layout of the framework experiments. We then investigate and evaluate our approach through a comprehensive evaluation and compare the results with various other approaches in indoor–outdoor detection.

### 5.1. Indoor–Outdoor Detection (IOD) Datasets

#### 5.1.1. Public CIOD Dataset

Cambridge indoor–outdoor detection [26] (CIOD) was obtained by collecting data from locations close to each other. This resulted in significant variations in signal patterns in different environments. The dataset includes feature sets collected over six months from smartphones, specifically the two Chinese Xiaomi Redmi Note 9 and Huawei P30 lite devices. The dataset consists of eight features: ambient light intensity, sound intensity, magnetic intensity, quality of Reference Signal Received Quality (RSRQ), proximity sensor readings, and a binary variable indicating day and night. These features provide information about daily activities such as work and travel. The data were recorded under weather conditions, including rain and clear sky. Min and max normalization operations were applied to normalize the sensor data with a sampling rate of 1 Hz. Before normalization, any significant outliers, identified as the top 1% of the sample distribution, were removed to ensure accurate data representation and to prevent any issues caused by malfunctioning smartphone sensors. The dataset size was more than 1.4 million samples for the training, while more than 14,000 samples were used for testing. Cellular signal strength plays a crucial role by indicating signal variations that mark transitions between indoor and outdoor environments. WiFi signal strength contributes by assessing the quality and intensity of nearby WiFi networks, which differ significantly between indoor and outdoor settings. Ambient light intensity is another important sensor, helping to distinguish environments based on light levels. The accelerometer detects user motion patterns, providing insights into whether a person is inside or outside. Total magnetic intensity offers insights into magnetic field variations, indicating different environments. Sound intensity helps differentiate environments based on noise levels. The proximity sensor ensures the reliability of ambient light data by confirming the presence or absence of nearby objects. Finally, the day/night label provides a temporal context for diurnal variations, enhancing the detection accuracy.

#### 5.1.2. Propose MIOD (Seen) Datasets

The mega indoor–outdoor detection (MIOD) datasets are carefully collected to classify indoor and outdoor environments. MIOD uses data from the Samsung Galaxy Fold, including accelerometer, magnetometer, and gyroscope sensors, sampled uniformly at 10 Hz within a 15 km radius around KAIST University. Its trajectory forms a closed-loop circular path with three stops representing different activity contexts. Stop proportions vary, with 10% allocated to starting and ending points, 25% to the indoor–outdoor transitions and 20% to predominantly indoor activities. The labeling follows established conventions, distinguishing between “Indoor” and “Outdoor” classes. Both datasets support the development of classification algorithms by offering precise geographical routes, strategic stops, and careful labeling, thereby contributing to the advancement of indoor–outdoor detection classification problems using only inertial sensors. The MIOD datasets include more than 1 million samples for training. In comparison, more than 14,000 samples are used for testing. The magnetometer is valuable for detecting changes in orientation and direction, offering insights into magnetic signatures that aid in distinguishing indoor and outdoor environments by analyzing magnetic field strength and direction variations. The accelerometer is essential for capturing movement patterns and velocity changes, helping to differentiate between indoor and outdoor spaces by identifying acceleration patterns indicative of transitions, such as sudden velocity changes or shifts in gravitational orientation. The gyroscope provides information about rotational movements and orientation changes, aiding in identifying transitions between indoor and outdoor environments by analyzing rotational patterns associated with specific activities or environmental changes.

#### 5.1.3. Propose DIOD Datasets (Unseen) Dataset

We also proposed Deep Indoor Outdoor Detection (DIOD) as an unseen dataset for a comprehensive evaluation after training in the MIOD dataset. Like MIOD, DIOD is collected using Samsung Galaxy Fold device sourced from South Korea, and sampled uniformly at 10 Hz at six various locations within KAIST-like Indoor Corridors, Indoor Hallways, Indoor Stairs, Outdoor Campuses, Parking lots, and Outdoor Roads. Overall, 50% of the samples were collected for indoor environments, with 20% each for Indoor Corridor and Indoor Hallway, and 10% for Indoor Stairs. The remaining 50% of the samples were collected for the outdoor environments: 25% for the Outdoor Campus, 10% for the Parking Lot, and 15% for the Outdoor Roads. The dataset included more than 1.4 million samples for training, while more than 14000 samples were used for testing.

### 5.2. Data Pre-Processing

Data are converted into numerical forms during the pre-processing stage via one-hot encoding. This method allows categorical variables to be represented as binary vectors, thus facilitating their integration into machine learning algorithms. Subsequently, the encoded categorical features are meticulously analyzed to assess the model’s robustness against perturbations and uncertainties in data representation. To address the issue of missing data, a systematic approach is taken to manage absent values in the dataset. Initially, feature columns consistently lacking data across observations are identified and excluded from further analysis. Following this step, imputation strategies are employed to address missing values within both categorical and numerical attributes. Specifically, for categorical features, missing entries are inputted using the mode corresponding to the most frequently occurring category in the entire dataset. In contrast, numerical features with missing values are inputted using the mean of the available data. Applying these pre-processing techniques prepares the dataset for further analysis, ensuring data integrity and reliability for modeling purposes.

### 5.3. DeepIOD Neural Network Implementation

The proposed models are designed for Indoor and Outdoor Detection Classification using Ensemble Learning with six cross-validation folds. The model architecture uses Keras, which can run as a top-level wrapper of the TensorFlow framework. TensorFlow was the backend for neural network training and inference in this experiment. The hardware used for the experiments consisted of a PC equipped with an Intel i9 CPU sourced from South Korea, operating at 3.20 GHz, 32 GB of DDR5 random access memory, and an RTX 4090 GPU. The proposed neural network is trained in a supervised fashion, and the gradient is back-propagated from the softmax activation layer to the LSTM layer. Randomly selected weights and biases are applied. To speculate the error between the model’s estimated and ground truth values, the cross-entropy loss function is used. The model is trained for 100 epochs with early stopping, setting the batch size to 64, with a learning rate of 0.001 and Adam [38] as an optimizer for stable convergence. The training set is randomly shuffled to increase the model’s robustness during training. Finally, using TensorFlow lite, lite models are generated for Android and resource-constrained device implementations.

### 5.4. Real-Time Experimental Scenarios

We conducted major experiments in three different countries for robust evaluation—Daejeon, Republic of Korea; Changzhou, China; and Lahore, Pakistan—for thorough evaluations of our framework, as shown in Figure 6.

### 5.5. Performance Evaluation

In this section, we present the performance of the DeepIOD system. We first analyze the parameters used for model training and evaluate the effect of each component in DeepIOD. Then, we compare DeepIOD with traditional indoor–outdoor detection systems. Thereafter, we comprehensively evaluate our model, considering each environment and detection instance. The robustness of our model is evaluated by recognizing different indoor and outdoor scenarios involving new participants. Finally, the real-time performance of DeepIOD is discussed.

### 5.6. Comparison with Related IOD Works

In Table 5, we present a comprehensive performance analysis of various indoor–outdoor detection methods evaluated in the CIOD public dataset. The comparison metrics include Accuracy, F1-Score, Precision, and Recall. SenseIO and IODetector, earlier methods from 2018 and 2012, respectively, show moderate performance, with accuracies of around 67.1% and 68.1%. The Random Forest (RF) method significantly improves these metrics, with an accuracy of 85.59%, reflecting advancements in machine learning approaches. Multi-Layer Perceptron (MLP) and Dense-LSTM models, further utilizing deep learning techniques, show enhanced performance, with accuracies of 86.98% and 88.05%, respectively. CAP-ALSTM, a more recent approach from 2022, achieves an accuracy of 89.36%, indicating the effectiveness of combining attention mechanisms with LSTM networks. The MB-SVM-HMM method, integrating Support Vector Machines and Hidden Markov Models, achieves the highest accuracy among related works at 92.17%. The proposed models (Model A, B, and C) outperform all related works, with Model B achieving the highest accuracy of 95.32%, followed by Model A at 94.17% and Model C at 93.76%. These results highlight the superior performance and robustness of the proposed models, particularly Model B, which excels in precision and recall as well, demonstrating the effectiveness of the new approaches in indoor–outdoor detection tasks; the superiority of our models is shown in Figure 7.

### 5.7. Comparison of Trained Models’ Accuracy on the Seen Dataset and Unseen Dataset

Table 6 shows that we evaluated the proposed models (Models A, B, and C) on both seen and unseen datasets to assess their robustness and generalization capabilities. The models were trained on the MIOD dataset (seen dataset) and tested on both MIOD and DIOD (unseen dataset). When tested on the seen dataset, all models exhibited high accuracy, with Model B achieving the highest accuracy of 94.28%, followed closely by Model A at 93.58% and Model C at 92.88%. These high accuracies indicate that the models effectively learned from the training data. However, when tested on the unseen DIOD dataset, there was a noticeable drop in performance, which is expected due to the different data distributions in the unseen dataset. Despite this drop, the models still demonstrated reasonable accuracy, with Model C showing the best generalization, with an accuracy of 77.79%, followed by Model B at 76.02% and Model A at 75.82%. This evaluation highlights the models’ ability to maintain a significant level of performance even when encountering new and unseen data, with Model C particularly excelling in adapting to the unseen dataset.

### 5.8. Comprehensive Evaluation of the DeepIOD Framework on Different Locations

We present the accuracy of detecting indoor–outdoor transitions, as shown in Table 7 and shown in Figure 8 in three countries: Korea, China, and Pakistan. The results indicate the robustness and effectiveness of the context-aware detection method. In Korea, the accuracy of the detection of the transition between indoors and outdoors is 97.21%, with a standard deviation of ±2.23, showcasing high reliability and consistency. In China, the accuracy is slightly higher at 97.78%, with a lower standard deviation of ±1.52, indicating a more stable performance. In Pakistan, the detection accuracy is 96.97%, with a standard deviation of ±2.47, which, while slightly lower than Korea and China, still represents a very high level of accuracy. These results demonstrate the method’s capability of maintaining high accuracy across different environmental contexts and geographic locations, underscoring its generalizability and adaptability to various settings.

In addition, we performed a comprehensive evaluation of a context-awareness-based DeepIOD framework in different environment settings, as shown in Table 8 and shown in Figure 9, which provide an in-depth assessment of the DeepIOD app’s performance across various indoor and outdoor environments in Korea, China, and Pakistan. The evaluations are based on different sensor combinations, including IMU (Magnetometer, Accelerometer, and Gyroscope), light, and GNSS sensors. For indoor environments, the app demonstrated high accuracy in all settings. In the indoor corridor environment, China recorded the highest accuracy at 98.44%, with a standard deviation of ±1.45, followed by Pakistan at 98.00% (±2.09) and Korea at 97.29% (±1.73). In the indoor hallway setting, Korea led with 97.74% (±1.15), while China and Pakistan followed closely at 96.85% (±1.45) and 97.51% (±1.60), respectively. For indoor stairs, China again showed superior performance, with 98.27% (±2.44), with Korea and Pakistan achieving 96.66% (±1.78) and 97.74% (±1.27), respectively. In outdoor environments, the app also performed exceptionally well. For outdoor open spaces, Pakistan achieved the highest accuracy at 98.48% (±0.96), followed by Korea at 97.96% (±1.90) and China at 96.58% (±2.05). In outdoor parking areas, Korea led with 98.20% (±2.09), while China and Pakistan followed at 96.96% (±1.21) and 96.90% (±1.00), respectively. For outdoor roads, Pakistan again showed high performance at 97.59% (±2.49), with Korea at 96.91% (±2.77) and China at 97.15% (±2.80). The results indicate that the DeepIOD app consistently achieves high accuracy across diverse environments and geographic locations. The slightly higher performance in some settings suggests that certain environmental factors or sensor integrations may contribute to better detection capabilities in those areas. This comprehensive evaluation underscores the app’s robustness and adaptability in real-world applications.

### 5.9. Majority Voter Comparison with Ground Truth

The comparison of the majority voter approach with the ground truth is demonstrated in Figure 10, which illustrates the performance of various input sensors, including IMU sensors, the light sensor, and GNSS, in detecting indoor and outdoor environments over time. The diagram features five rows, each representing a distinct aspect of the detection process. The top row (blue line) presents the predictions from the Deep Neural Network (DNN) models using IMU sensors, specifically the magnetometer, accelerometer, and gyroscope. The alternating high and low states of this line indicate transitions between indoor and outdoor environments. The second row (orange line) displays the predictions from the light sensor, which also fluctuate, capturing changes in lighting conditions associated with moving between indoor and outdoor spaces. The third row (green line) shows the predictions from the GNSS sensor, with transitions reflecting the sensor’s ability to detect outdoor environments, typically characterized by better GNSS signal reception. The fourth row (red dashed line) represents the output of the majority voter algorithm, which integrates the predictions from the IMU, light, and GNSS sensors. This majority voter output aims to provide a more accurate and robust detection by considering the input consensus. Finally, the bottom row (black dotted line) depicts the ground truth, representing the actual indoor–outdoor transitions during the test period. This is the benchmark against which the sensor predictions and the majority voter output are compared.

Table 9 compares the trained models’ accuracy on seen and unseen datasets using only IMU sensors, incorporating the percentage improvement for each model relative to the DeepIOD framework average accuracy of 97.32%. The percentage improvement, calculated as the relative increase in accuracy, is detailed alongside each model, providing a comprehensive assessment of performance enhancements.

## 6. Conclusions

Indoor–outdoor detection (IOD) has gained significant attention because of its crucial role in positioning technologies and environmental change detection using multimodal smartphone sensors. This study reviewed the current state of IOD, focusing on deploying location-based services in embedded systems that utilize low power consumption and on-device artificial intelligence. The main objective of the proposed framework is to accurately classify environments as indoor or outdoor by integrating sensory data, location information, adaptive threshold mechanisms for light intensity and GPS, and social choice mechanisms for combining decisions from deep learning models. The DeepIOD framework integrates IMU sensor data, GPS, and light sensors, preprocesses these data, and uses multiple deep neural network models and sensor modules to robustly predict whether the environment is indoor or outdoor. Extensive experiments conducted on six unseen environments using a smartphone and TensorFlow Lite packages demonstrated the efficacy of this approach, with accuracy rates ranging from 98% to 99%. These results surpass existing methods based on thresholding, traditional machine learning, and shallow/deep learning techniques. The findings of this study highlight the superiority of the DeepIOD method over existing methods, paving the way for more reliable and efficient IOD in smart IoT environments.

## Figures and Tables

**Figure 1 sensors-24-05125-f001:**
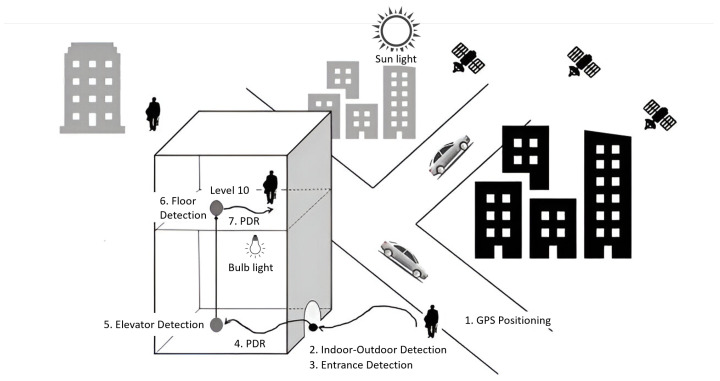
Indoor–outdoor-integrated GPS system [2].

**Figure 2 sensors-24-05125-f002:**
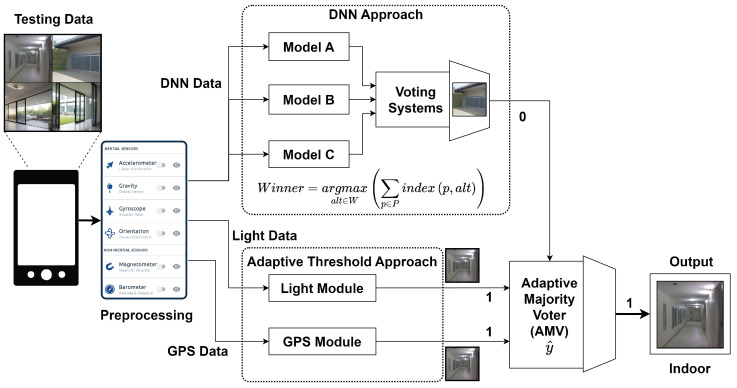
DeepIOD framework for indoor–outdoor environment classification.

**Figure 3 sensors-24-05125-f003:**
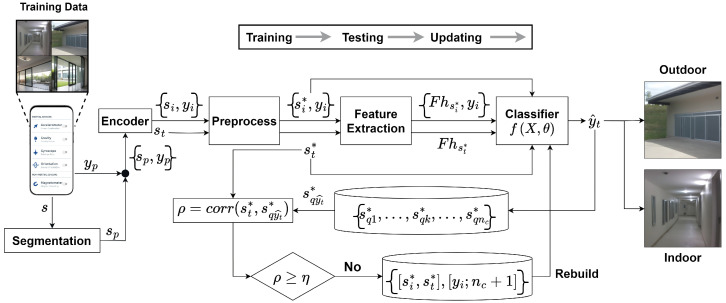
DeepIOD proposed DNN downstream.

**Figure 4 sensors-24-05125-f004:**
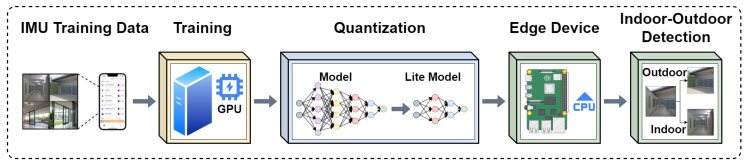
Conversion of DNN models into lite version.

**Figure 5 sensors-24-05125-f005:**
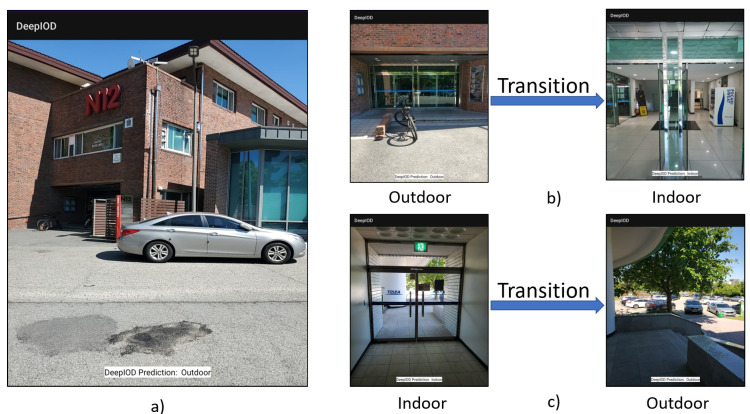
Here, (**a**) shows the DeepIOD smartphone app, (**b**) shows the transition from outdoor to indoor, and (**c**) shows the transition from indoor to outdoor.

**Figure 6 sensors-24-05125-f006:**
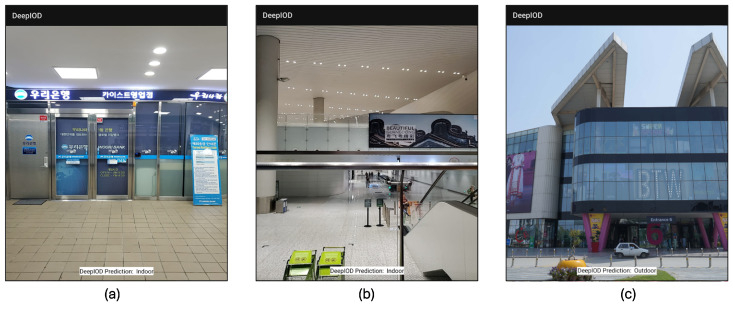
DeepIOD application experiment at three different locations: (**a**) night, indoor, Woori bank KAIST branch, in Daejeon, Republic of Korea; (**b**) night, indoor, Changzhou, China; and (**c**) day, outdoor, Lahore, Pakistan.

**Figure 7 sensors-24-05125-f007:**
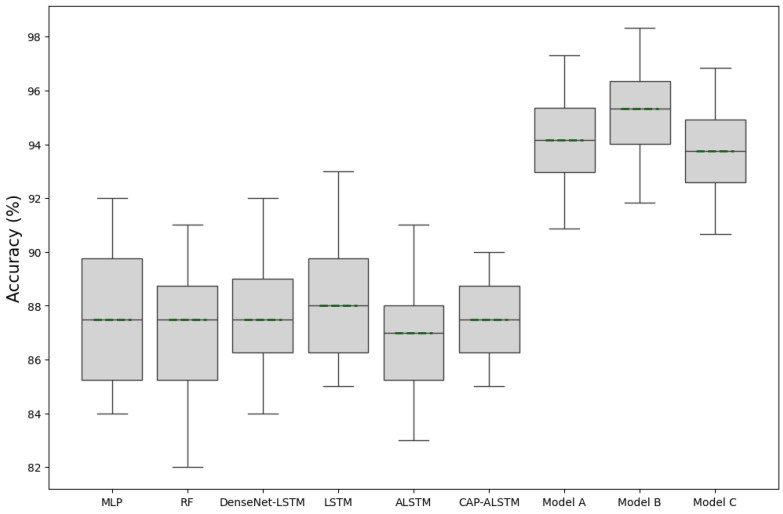
Model comparison including proposed models (A, B, C).

**Figure 8 sensors-24-05125-f008:**
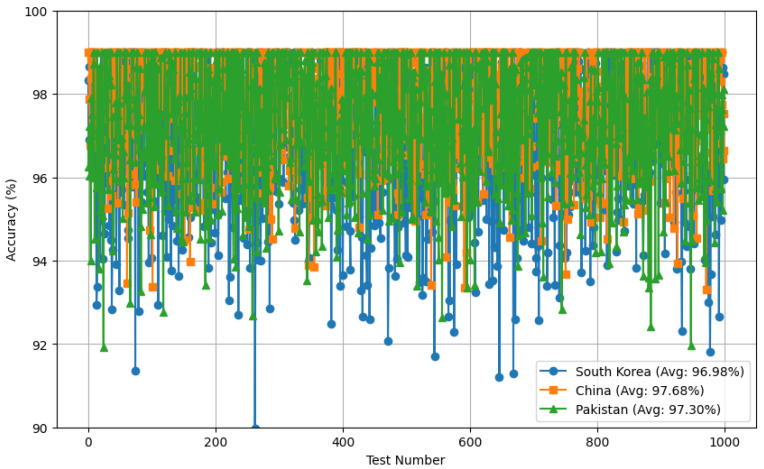
Indoor–outdoor transition detection accuracy over 1000 tests.

**Figure 9 sensors-24-05125-f009:**
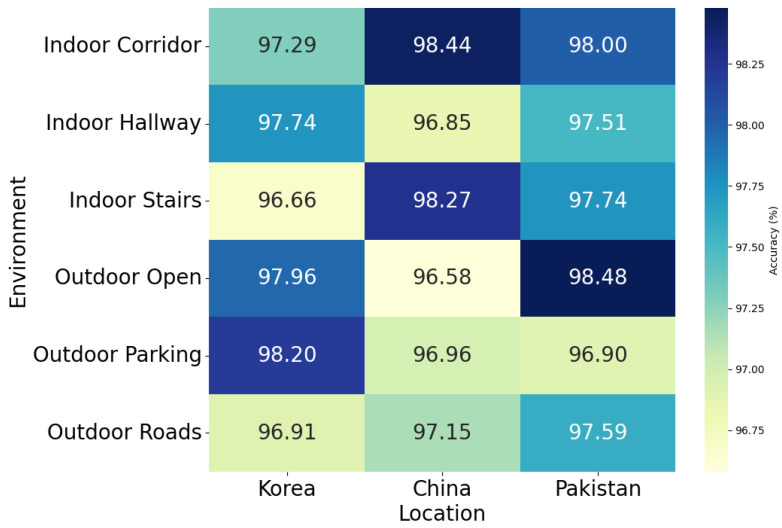
Heatmap of the DeepIOD app experimental results at different environments with regard to location.

**Figure 10 sensors-24-05125-f010:**
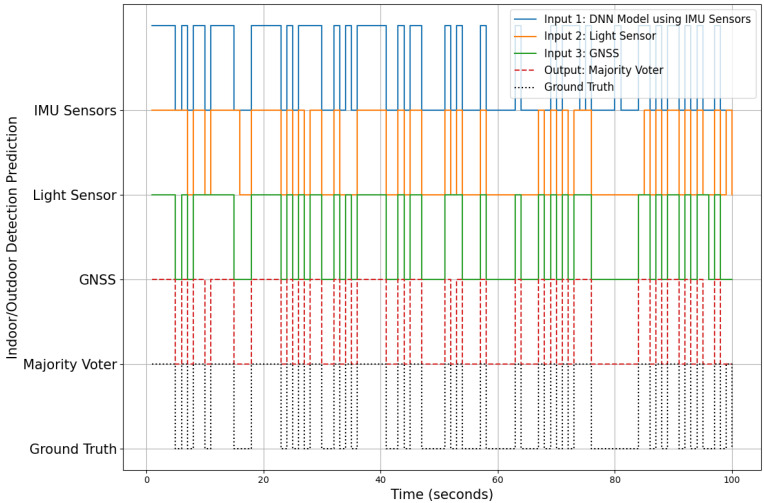
Majority voter comparison with ground truth for indoor–outdoor detection.

**Table 1 sensors-24-05125-t001:** Architecture of models A, B, and C.

Component	Model A	Model B	Model C
Input	Input Layer	Input Layer	Input Layer
Layer 1	Conv1D (64, k=3)	Conv1D (64, k=3)	DWS-Conv2D (3×3)
Layer 2	MaxPool (2)	LayerNorm + MHA (8, 64)	Conv2D (32, 1×1)
Layer 3	LSTM (64)	MaxPool (2)	DWS-Conv2D (3×3)
Layer 4	Dropout (0.2)	GlobalAvgPool	Conv2D (32, 1×1)
Layer 5	Dense	Reshape + Dense	MaxPool (2×1)
Output	Dense		

**Table 2 sensors-24-05125-t002:** Comparison of traditional voting methods and proposed voting classifier.

Voting Method	Limitations	Proposed Improvement
Plurality	Vulnerable to vote splitting. Does not ensure the selection of the most preferred option.	Consider pairwise comparisons and overall rankings.
STV ^1^	Complex ballot counting process. Vulnerable to strategic voting.	Simplifies the decision-making process. Considers preferences across multiple options.
Condorcet	May not always produce a winner. Vulnerable to cycling.	Systematically addresses tie-breaking.
Borda	Susceptible to strategic manipulation. May not select the most preferred option.	Scoring based on pairwise comparisons.
Copeland	Complexity in determining the winner. Vulnerable to strategic voting.	Simplifies the selection process. Considers the preferences of voters.
Dictatorship	Concentration of power. Lack of representation for minority views.	Based on collective preferences of multiple classifiers.

^1^ Single Transferable Vote.

**Table 3 sensors-24-05125-t003:** Light intensity thresholds for different locations.

Location	Indoor Light Intensity Threshold (lux)	Outdoor Light Intensity Threshold (lux)	Adaptation Mechanism
Republic of Korea	100–300	1000–20,000	Continuously adapts based on weather changes, seasonal daylight variations, and urban light pollution.
China	150–350	1200–25,000	Adjusts to account for high pollution levels affecting natural light and diverse climatic conditions.
Pakistan	80–250	800–18,000	Adapts considering frequent power outages affecting indoor lighting and strong seasonal daylight shifts.

**Table 4 sensors-24-05125-t004:** Fixed threshold for GPS horizontal accuracy.

Classification	Horizontal Accuracy (meters)
Indoor	>5
Outdoor	<5

**Table 5 sensors-24-05125-t005:** Comparison of our proposed three models with related works on public dataset CIOD ^1^ [26].

Method	Accuracy (%)	F1-Score (%)	Precision (%)	Recall (%)
SenseIO [21]	67.1±5.80	77.8±5	77.6±3.0	77.4±5.3
IODetector [20]	68.10±8.47	77.7±6.79	77.9±3.5	77.8±5.37
RF [32,39]	85.59±8.42	87.75±7.6	85.2±4.5	87.75±6.2
MLP [30,39]	86.98±6.50	89.14±5.8	84.3±3.7	89.14±5.0
Dense-LSTM [24]	88.05±6.42	89.84±5.56	87.5±2.6	89.84±4.0
CAP-ALSTM [26]	89.36±5.28	90.97±5.06	89.8±3.1	90.97±4.5
MB-SVM-HMM [23]	92.17±2.23	92.35±2.46	91.7±1.9	92.35±2.1
**Model A ^2^**	94.17±3.13	94.65±3.62	93.78±2.19	94.35±3.69
**Model B ^2^**	95.32±3.48	94.57±3.41	95.27±3.21	95.28±3.34
**Model C ^2^**	93.76±3.09	93.07±3.59	93.23±3.14	93.68±3.82

^1^ Seen dataset and ^2^ Proposed DeepIOD models.

**Table 6 sensors-24-05125-t006:** Comparison of trained models accuracy on seen and unseen datasets.

Training Dataset	Testing Dataset	Model A (%)	Model B (%)	Model C (%)
MIOD ^1^	MIOD ^1^	93.58±3.48	94.28±2.32	92.88±2.87
MIOD ^1^	DIOD ^2^	75.82±3.89	76.02±2.91	77.79±1.87

^1^ Seen dataset and ^2^ unseen dataset.

**Table 7 sensors-24-05125-t007:** Context-awareness-based indoor–outdoor transition detection accuracy.

Environment	Korea	China	Pakistan
IO Transition ^1^	97.21±2.23	97.78±1.52	96.97±2.47

^1^ Indoor–outdoor transition detection.

**Table 8 sensors-24-05125-t008:** Comprehensive evaluation of context-awareness-based DeepIOD app experiments in different environmental settings.

Environment	Korea (%)	China (%)	Pakistan (%)	Sensors ^1^
Indoor Corridor	97.29±1.73	98.44±1.45	98.00±2.09	IMU+L+G
Indoor Hallway	97.74±1.15	96.85±1.45	97.51±1.60	IMU+L+G
Indoor Stairs	96.66±1.78	98.27±2.44	97.74±1.27	IMU+L+G
Outdoor Open	97.96±1.90	96.58±2.05	98.48±0.96	IMU+L+G
Outdoor Parking	98.20±2.09	96.96±1.21	96.90±1.00	IMU+L+G
Outdoor Roads	96.91±2.77	97.15±2.80	97.59±2.49	IMU+L+G

^1^ IMU sensors = Magnetometer, Accelerometer and Gyroscope, L = light, G = GNSS.

**Table 9 sensors-24-05125-t009:** Comparison of percentage improvement of DeepIOD with IMU sensors only.

Method	Accuracy (%)	Average Accuracy ^1^ (%)	Percentage Improvement
Model A ^2^ (IMU)	93.58±3.48	97.32	3.99%
Model B ^2^ (IMU)	94.28±2.32	97.32	3.22%
Model C ^2^ (IMU)	92.88±2.87	97.32	4.78%
Model A ^3^ (IMU)	75.82±3.89	97.32	28.31%
Model B ^3^ (IMU)	76.02±2.91	97.32	28.00%
Model C ^3^ (IMU)	77.79±1.87	97.32	25.09%

^1^ DeepIOD framework, ^2^ seen dataset, ^3^ unseen dataset.

## Data Availability

Data are contained within the article.

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
