# Peer review of "DeepIOD: Towards A Context-Aware Indoor–Outdoor Detection Framework Using Smartphone Sensors"

_sensors, 2024, doi:10.3390/s24165125_

Round 1

Reviewer 1 Report

Comments and Suggestions for Authors

The manuscript entitled" DeepIOD: Towards Context-Aware Indoor-Outdoor Detection Framework using Smartphone Sensors" describes the enhancement of the failure in indoor environments due to signal obstructions. This was achieved by introducing the DeepIOD framework, which leverages IMU sensor data, GPS, and light information. This manuscript presented an important solution to the Indoor-Outdoor Detection system. It requires minor revisions as follows:

1.  The abstract doesn’t contain a clear description of the problem. Also, more modifications are required to improve the abstract by adding important results.

2.  The introduction section needed more details to present the previous work of different authors applied to DeepIOD.

3.  Also, please remove the results presented at the end of the introduction section (line 71 on page 2) "Our DeepIOD system achieves a remarkable accuracy rate of 98%-99% with a transition time of less than 10 ms".

4.  The novelty of the present study must be modified to be clear for readers.

5.  On page 7, in the methodology section, more details are needed to explain Figure 3. which describes the DeepIOD of DNN downstream.

6.  More discussions are needed for Figures (8) and Table (4) in the results and discussion sections. A comparison of the present work results with the results of other works is also needed.

Best regards for authors

Reviewer 2 Report

Comments and Suggestions for Authors

The study proposes the DeepIOD framework for accurate indoor-outdoor detection using smartphone sensors. It integrates data from IMU sensors, GPS, and light sensors, preprocesses this data, and employs multiple deep neural network models and an adaptive majority voting mechanism to classify environments as indoor or outdoor. 

Comments:

1. How does the adaptive majority voting mechanism improve the accuracy of the IOD classification compared to traditional voting methods?

4. What preprocessing steps are applied to the sensor data before feeding it into the deep learning models?

5. How is the threshold for light intensity adaptively adjusted in the framework?

8. What is the computational complexity of the proposed framework, and how does it affect real-time performance?

9. How is the DeepIOD app evaluated in real-time scenarios, and what are the results?

1. Clarify the novelty of the proposed framework compared to existing IOD methods.

5. Address the grammatical errors and typos throughout the manuscript.

Reviewer 3 Report

Comments and Suggestions for Authors

This study tried to propose a DeepIOD framework, which leverages IMU sensor data, GPS, and light information to accurately classify indoor and outdoor environments. The framework preprocesses the input data, employs multiple deep neural network models, and combines the outputs using an adaptive majority voting mechanism. Experiments using a smartphone in six unseen environments show that DeepIOD achieves significantly higher accuracy than methods using only IMU sensors. However, based on the current form, the manuscript still requires further clarification and revision, and the author should consider the following comments.

Major comments,

1. The theoretical part lacks a clear and detailed description, and there are many omissions and jumps in the formulas. As a reader, I find it difficult to understand the theoretical part. Detailed explanations and further clarifications should be provided for this part.

2. This paper integrates a variety of data, but lacks a detailed description of how the multi-source data are combined and how the quality control is carried out. Data quality control and the fusion of multi-source data are essential steps, but this paper lacks a description in this regard and supplementary explanations should be provided.

3. When calculating on the phone side, the calculation efficiency is also very important, but it is not mentioned in the manuscript. What is the training cost (time) of the neural network? How applicable it is in different environments should be further explained.

4. This paper uses three types of data as input. Then, what are the roles of the three types of data respectively, and how to use these three types of data? Why are there differences in the results in different regions? Is it due to the performance difference of GPS results or other factors?

Minor comments,

1. Eq (1), where is sensory data D and location information L in this equation? how to use these data? N refer to what? how to understand this equation?

2. Eq (2), it is difficult to understand why this equation presented here.

3. For all equations, you should explain all parameters in the equation.

4. Eq (7), I still cannot understand how to calculate Threshold in equation 7.

5. Figure 2, the model applied should be presented. which model are used in DNN approach? it can be automatic selected? According to what criterion?

6. How the weight of data changes from indoor to outdoor, how the data input information changes, and how the neural network in this paper adaptively adjusts the parameters, these most crucial issues should all be described in detail.

7. Figure 7, the definition of accuracy should be given. And how to calculate this value? According to the previous theories, all the three models A, B, and C are in the neural network, then why are they assessed individually here, rather than within a unified model? The information of all schemes listed in figure 7 should be described in detail.

8. Table 1 and 2, the unit of accuracy?

9. How does the proposed scheme in this study compare with the existing schemes? For the neural network training in this paper, what are the input and output respectively, and how to establish the training group?

Comments on the Quality of English Language

The language expression in this paper is clear.

Reviewer 4 Report

Comments and Suggestions for Authors

The introduced DeepIOD framework is innovative, and the composition of the deep neural network models is detailed in the paper, with experimental validation conducted in six scenarios. There are two minor questions for the authors' reference:

 1. Is the prediction result of Fig. 5b, where all results are marked as outdoor, a writing error?

2. Compared to the method using only IMU sensors, could you quantitatively calculate how much the accuracy of the DeepIOD framework has improved?

Round 2

Reviewer 2 Report

Comments and Suggestions for Authors

The authors addressed my concerns.

Author Response

The paper has been thoroughly revised.

Reviewer 3 Report

Comments and Suggestions for Authors

After the last round of revisions, the author has basically solved my confusions and problems. However, for the current GNSS mobile phone navigation and positioning, multi-system has become the norm. How to consider the intervention of multiple signals? Besides, all the formulas should be checked in detail to ensure that all the symbols are explained. Why is a fixed threshold of 5 meters set for GPS? How is this value considered? Finally, and most importantly for this paper, the introduction of the neural network algorithm. For this "black box" algorithm, this paper still fails to explain and clarify it well, although the author has spent a lot of space explaining it. On the whole, this paper presents a multi-source fusion algorithm for indoor and outdoor positioning on mobile phones and can be published after appropriate revisions and clarifications.

Author Response

Comment 1:How to consider the intervention of multiple signals? Besides, all the formulas should be checked in detail to ensure that all the symbols are explained.

Response 1: The query has been addressed in the introductory paragraph of section 4. The paragraph is revised.

Comment 2: Why is a fixed threshold of 5 meters set for GPS? How is this value considered?

Response 2: The query has been addressed in section (4.5) with reference.

Comment 3: Finally, and most importantly for this paper, the introduction of the neural network algorithm. For this "black box" algorithm, this paper still fails to explain and clarify it well, although the author has spent a lot of space explaining it.

Response 3: The query has been thoroughly addressed in section (4.2). The Table (1) and mathematical equations have been added.